# DynamicsDiffusion: Generating and Rare Event Sampling of Molecular Dynamic Trajectories Using Diffusion Models

## Abstract

Molecular dynamics simulations are fundamental tools for quantitative molecular sciences. However, these simulations are computationally demanding and often struggle to sample rare events crucial for understanding spontaneous organization and reconfiguration in complex systems. To improve general speed and the ability to sample rare events in a directed fashion, we propose a method called *Dynamics-Diffusion* based on denoising diffusion probabilistic models (DDPM) to generate molecular dynamics trajectories from noise. The generative model can then serve as a surrogate to sample rare events. We leverage the properties of DDPMs, such as conditional generation, the ability to generate variations of trajectories, and those with certain conditions, such as crossing from one state to another, using the 'inpainting' property of DDPMs, which became only applicable when generating whole trajectories and not just individual conformations. To our knowledge, this is the first deep generative modeling for generating molecular dynamics trajectories. We hope this work will motivate a new generation of generative modeling for the study of molecular dynamics.

## 1 Introduction

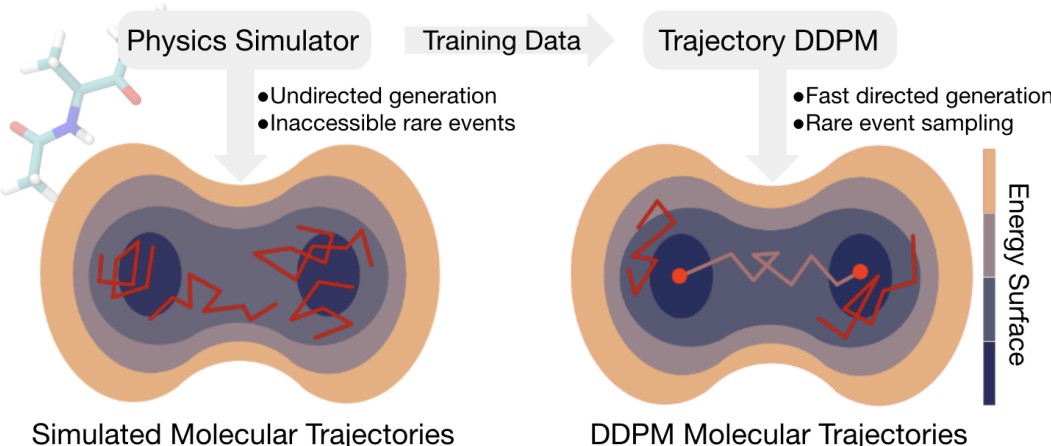

Figure 1: DynamicsDiffusion is trained on molecular dynamics trajectories generated via the numerical integration of Newton's laws of motion for a molecular system. The trained diffusion model is then a surrogate of the simulator but with added properties of speed and steerability. This steerability allows for the sampling of rare events, like state transitions, that are crucial to the understanding of molecular systems but are often not or rarely observed in conventional simulations.

Diffusion models Sohl-Dickstein et al.Ho et al. (a)Song et al. (c) have recently emerged as a powerful approach to generative modeling, allowing for significant advances in the field of image synthesis

Ramesh et al.Rombach et al.. These models can produce high-quality samples that capture the complex distributions of natural images. Early attempts to apply these models to dynamics are being undertaken for video Ho et al. (b) and action modeling Janner et al.Chi et al., with promising results. In the context of the molecular sciences, diffusion models have so far been used to generate ensembles of molecular conformations conditioned to thermodynamic parameters Wang et al.Arts et al.. However, their potential in the study of the dynamics of molecular systems is still unexplored. Deep learning has made outstanding contributions in the structural modeling of molecular systems Jumper et al., However, an accurate description of the dynamics is essential to understand the biological function of proteins and other molecules. Here, we make a first attempt to extend diffusion models from the generation of conformations to the whole trajectories encoding the molecule's dynamics. We show how we can take advantage of a variety of guided generation methods that allow for the targeted sampling of the dynamics of otherwise rare but important events.

Molecular dynamics (MD) simulations are physics-based simulations where the system propagates in time according to Newton's second law of dynamics on a specified energy function Dror et al.. These simulations are one of the standard methods to obtain quantitative insight into the dynamic organization of complex molecular systems. To guarantee physical accuracy and numerical stability, the integration of the dynamics must proceed in very small iterations. For instance, standard classical atomistic MD must use an integration time step of 2 fs. The disparity between the integration time step and the time scales on which interesting biomolecular events occur severely limits the potential of classical MD simulations. This limitation is particularly severe for "rare events"—many conformational reorganizations, like large conformational motions, folding, and assembly, but also binding and more general collective rearrangements— which occur on time scales that are many orders of magnitude larger than the typical integration time steps.

Machine learning, particularly deep learning, has emerged as a promising approach to speed up and guide the sampling in simulations. Two common strategies are to either approximate the physical forces and enable larger integration timesteps by using Graph Neural Networks (GNN) Park et al.Li et al. (b), or to learn a coarse-grained representation of the system and its dynamics to speed up the simulation, with GNN's Husic et al.Li et al. (a), or by using diffusion models Arts et al..

The guided sampling of rare events is particularly useful for systems with hindered ergodicity, i.e., where large energy barriers separate alternative metastable states. For some systems, these events occur so rarely that speeding up simulations is not sufficient. Therefore, new methodological approaches are essential to focus computational resources on sampling trajectories where the system overcomes the energy barriers. Umbrella sampling Torrie & Valleau and metadynamics Laio & Parrinello are popular techniques to compute the thermodynamic properties of a system by introducing external biases to force the system to sample low-probability regions of its configuration space. A popular method, transition path sampling (TPS) Bolhuis et al. is used to study the kinetics of rare events by generating a set of unbiased reactive trajectories using a trajectory sampling scheme. TPS and related methods Jung et al. focus the sampling on transition paths—the short segments of trajectories that overcome barriers and connect different states.

The sampling of molecular conformations using deep generative models has been explored using normalizing flow methods like Boltzmann generators Noé et al. and recently diffusion models Wang et al.Arts et al.. The latter has come to the forefront of research due to their less strict requirements for the architecture and the ease of training even on high dimensional data Song & Ermon. Recently these models have been employed to sample conformations conditional on properties like temperature Wang et al. or reaction coordinates Falkner et al. in order to sample conformations of interest, including in low-probability regions of the molecular configuration space.

Here, we present an approach that builds on diffusion models to generate whole dynamic trajectories of molecular systems. Diffusion models have been successfully applied to learn time series data, in the form of action planning Janner et al.Chi et al., and molecular conformations Arts et al.Wang et al.. In essence, we train a surrogate of the physical integrator using a diffusion-based generative model that can perform enhanced sampling and generate trajectories of interest. This combines two promising research directions, both learning the integrator and sampling rare events using deep learning. Our approach presents two very attractive features. First, it enables us to speed up the sampling of the system's dynamics. Second, diffusion models offer ways of controlling the sampling absent from classical physical simulators.

We can take advantage of these properties to generate trajectories conditionally. Once our surrogate diffusion model is trained, we can: 1. Generate trajectories that are conditioned to a global parameter, like temperature; 2. Generate trajectories crossing the barrier between states by leveraging the 'inpainting' ability of diffusion modelsSong et al. (c)Song et al. (b)Chung et al.Lugmayr et al.. By fixing the start and end points of the trajectory in distinct states and allowing the model to fill in the transition, rare-event enhanced sampling becomes possible. 3. Generate an ensemble of (reactive) trajectories by partially noising and denoising them.

## 1.1 CONTRIBUTIONS

- We demonstrate the ability of diffusion models to learn to generate molecular dynamics trajectories and propose an architecture that is suitable for the symmetries and properties of this specific task.

- We show how this approach enables rare event enhanced sampling, the generation of variations in trajectories, the ability to generate transition paths between states, and conditional generation.

- We test the model by training it on a molecular system and show that it is able to reconstruct free energy and dynamics accurately.

## 2 DIFFUSION GENERATIVE MODELLING FOR TRAJECTORIES

### 2.1 MOLECULAR DYNAMIC TRAJECTORIES AS A DATATYPE

In this section, we introduce the notation for the training data. Let $\mathcal{D} = \{x_i\}_{i=1}^{K}$ represent the dataset of trajectory snippets sampling an unknown (non-normalized) probability density $p(x)$, where $x_i$ denotes the $i$-th training sample. Each sample is a trajectory of $\tau$ time steps, where $\tau$ is sampled uniformly from the range $[\tau_{min}, \tau_{max}]$ to aid multi-length generation at inference. This is equivalent to the ordered set, $x_i = \{R_\mu^{(i)}\}_{\mu=1}^{\tau}$, where $R_\mu$ is a configuration of the system at time step $\mu$. For a molecular system consisting of $N$ atoms, we can define a configuration as the set of Cartesian coordinates for each atom in the molecule, $R_\mu \in \mathbb{R}^{N \times 3}$, with each entry corresponding to a coordinate of an atom.

To represent the training data as a batch array, with batch size $b$, with shape ($b$, $\tau$, $N \times 3$), we first organize the trajectory snippets into batches. Let $\mathcal{B} = \{\mathcal{D}_k\}_{k=1}^{b}$ denote a batch, where $\mathcal{D}_k = \{\boldsymbol{x}_i^{(k)}\}_{i=1}^{K}$ represents the $k$-th set of trajectory snippets.

Each trajectory snippet $\boldsymbol{x}_i^{(k)}$ consists of $\tau$ time steps with $N$ atoms and their respective three-dimensional coordinates, however, the model can also be trained for multiple snippet lengths, to allow for the generation of trajectories of multiple lengths. The complete data array for a batch $\mathcal{B}$ can then be represented as $A \in \mathbb{R}^{b \times \tau \times N \times 3}$. Each array entry corresponds to a specific trajectory snippet, time step, and Cartesian atomic coordinate. The final data can be treated equivalently to an image in image diffusion models where the time dimension corresponds to the image width, albeit of a very wide shape due to the long time dimension.

### 2.2 DIFFUSION MODEL THEORY DESIGN CHOICES

Diffusion-based generative models have been independently formulated multiple times under different theoretical frameworks Sohl-Dickstein et al., Song et al. (c) and Ho et al. (a). To illustrate the relationship of the trained diffusion model with the forces of a physical system, we will introduce the score-matching interpretation since it makes this relationship apparent and highlights how it can be used to introduce an inductive bias into the diffusion model's architecture. In this formulation, the model learns to approximate the score $\nabla_x \log(p(x))$ instead of learning the distribution of the data $p(x)$ directly, avoiding the calculation of the normalization constant of $p(x)$, which is usually intractable. The score can be seen as a vector pointing toward the dense regions of the data distributions. Given the score, we can sample data points using an iterative sampling scheme based on Langevin dynamics Song et al. (c):

$$x_{i+1} \leftarrow x_i + \epsilon \nabla_x \log(p(x)) + \sqrt{2\epsilon} z_i, i = 0, 1, ..., K \tag{1}$$

Here $z_i$ is noise sampled from $\mathcal{N}(0, I)$ and if $K \to \infty$ and $\epsilon \to 0$ we sample one $x_k$ from $p(x)$. However, the estimation of this score is inaccurate in regions of low density in the data and even if approximation was possible the sampling would be inhibited by local minima, making sampling impossible. To counteract this limitation the model is trained to estimate the score of the data distribution at different levels of added noise, and conditioning the model on the noise level Song & Ermon. Adding noise to the data makes the data density more uniform, allowing for a more accurate score estimation. When sampling with this conditioned model the Langevin dynamics become annealed, decreasing in noise level at each step, till a sample from the non-noised data distribution is obtained. After the introduction of the noise levels, this framework becomes functionally equivalent to the DDPM approach described in the appendix section 'The noising and de-noising process'.

**Design Choices** When understanding the diffusion model under this framework Arts et al. show that a model trained on conformations of physical systems, $R_\mu^{(i)}$ and their equivalents along the noising process $R_{\mu,t}^{(i)}$, approximates the force-field of that system at that noise level. At noise level $t = 0$ this corresponds to the forces of the non-noised conformations, i.e. the original physical system generating the training data.

$$Model(R_{\mu,t=0}^{(i)}, t = 0) \sim \nabla_{R_{\mu,t=0}^{(i)}} \log(p_0(R_{\mu,t=0}^{(i)})) = \frac{-\nabla V(R_{\mu,t=0}^{(i)})}{k_{\mathrm{B}}T} = \frac{F_{R_{\mu,t=0}^{(i)}}}{k_{\mathrm{B}}T} \tag{2}$$

This follows since the distribution of a physical system's states is given by the Boltzmann distribution:

$$p(R_\mu^{(i)}) \propto e^{-\frac{V(R_\mu^{(i)})}{k_{\mathrm{B}}T}} \tag{3}$$

Based on this interpretation the diffusion model corresponds to the forcefield of the system. Conservative forces in physical systems are such that the work done by a particle moving between two points is the same irrespective of the path taken. This property is not satisfied by a basic UNet architecture Ronneberger et al. which is commonly used for images. However, it can be built in as an inductive bias by computing the forces as the gradient of an energy function, $-\nabla V(R_\mu^{(i)}) = F_{R_\mu^{(i)}}$ Arts et al. per construction. Including this bias led to more accurate reconstructions of trajectories in our work (see ablations).

In this work, we extend the inductive bias from individual conformations $R_\mu^{(i)}$ Arts et al. to trajectories $\boldsymbol{x}_i$. We implemented this by outputting one channel per time step meaning the UNet part of the model transforms one batch $\mathcal{B}$ of shape $\mathbb{R}^{b \times \tau \times N \times 3} \to \mathbb{R}^{b \times \tau}$ which can be interpreted as outputting a singular energy value per conformation $V_{R_\mu^{(i)}}$. These energies then get summed to a singular trajectory-level energy $V_{x_{i,t}}$. From this energy, we can then recover the forces of the same shape as the input by computing the Jacobian with respect to the model's input $-\nabla V_{x_{i,t}}$ 2.

Based on the success of UNet models with 1D-convolutions for time-series generation in the context of action planning Janner et al., we chose a similar architecture for our study, which additionally included self-attention layers Vaswani et al.. For the embedding of additional conditioning, such as temperature, we used a small multilayer perceptron, MLP, adding the embedding in each of the ResNet blocks.

## 2.3 ENHANCED SAMPLING METHODOLOGY

There are two different approaches to enhanced sampling via Dynamics diffusion, one based on inpainting and the other one based on variation generation. The inpainting variant works by imposing a few selected frames that the diffusion model must include during the generation of a trajectory.

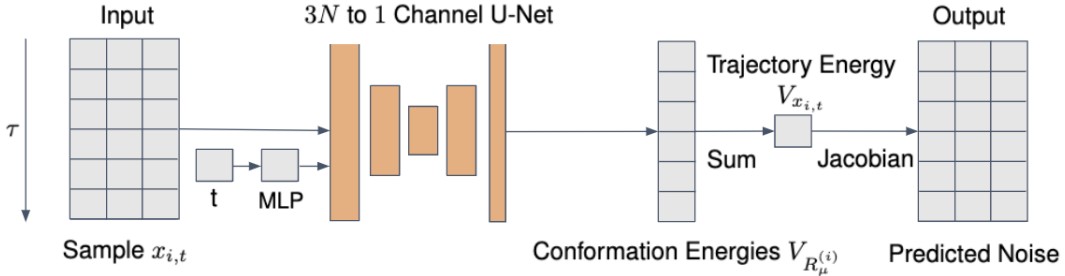

Figure 2: The model's architecture extends the conventional UNet to one with a physics-informed inductive bias.

The variation approach works by taking an existing trajectory (or pseudo-trajectory, like a linear interpolation), partially noising it, and then denoising it with the diffusion model, in this way creating variations of the original trajectory. How close to the original trajectory the variation should be can be determined by the amount of noise that is added before denoising it.

---

**Algorithm 1** Standard DDPM Sampling

---

1: $x_T \sim \mathcal{N}(0, I)$             ▷ Sample a noise vector
2: **for** $t = T, ..., 1$ **do**          ▷ Iterate through all noise levels t
3:     $z \sim \mathcal{N}(0, I)$ if $t > 1$, else $z = 0$     ▷ Sample a noise vector for the denoising operation
4:     $x_{t-1} = \frac{1}{\sqrt{\alpha_t}}(x_t - \frac{1-\alpha_t}{\sqrt{1-\alpha_t}})\epsilon(x_t, t) + \sigma_t z$     ▷ Perform one denoising step
5: **end for**
6: **Return** $x_0$

---

**Algorithm 2** Inpainting Generation

---

1: $x_T \sim \mathcal{N}(0, I)$             ▷ Sample a noise vector
2: select $x_{\text{fix}}$, $\text{index}_{\text{fix}}$      ▷ Select the frames that are imposed and their index/location
3: **for** $t = T, ..., 1$ **do**          ▷ Iterate through all noise levels t
4:     $z \sim \mathcal{N}(0, I)$ if $t > 1$, else $z = 0$ ▷ Sample a noise vector for the denoising operation and the noising of the inpainting condition
5:     $x_t[\text{index}_{\text{fix}}] = \sqrt{\alpha_t}x_{\text{fix}} + \sqrt{1 - \alpha_t}z$     ▷ Noise the imposed frames and insert them
6:     $x_{t-1} = \frac{1}{\sqrt{\alpha_t}}(x_t - \frac{1-\alpha_t}{\sqrt{1-\alpha_t}})\epsilon(x_t, t) + \sigma_t z$     ▷ Perform one denoising step
7: **end for**
8: **Return** $x_0$

---

**Algorithm 3** Variation Generation

---

1: $x_T \sim \mathcal{N}(0, I)$             ▷ Sample a noise vector
2: select $x_{\text{variation}}$          ▷ Select a sample to generate variations of
3: select $t_{\text{start}}$        ▷ Select a noise level to noise the variation sample to
4: $x_{t_{\text{start}}} = \sqrt{\alpha_{t_{\text{start}}}}x_{\text{variation}} + \sqrt{1 - \alpha_{t_{\text{start}}}}x_T$     ▷ Noise $x_{\text{variation}}$ to noise level $t_{\text{start}}$
5: **for** $t = t_{\text{start}}, ..., 1$ **do**       ▷ Iterate through all noise levels after $t_{\text{start}}$
6:     $z \sim \mathcal{N}(0, I)$ if $t > 1$, else $z = 0$     ▷ Sample a noise vector for the denoising operation
7:     $x_{t-1} = \frac{1}{\sqrt{\alpha_t}}(x_t - \frac{1-\alpha_t}{\sqrt{1-\alpha_t}})\epsilon(x_t, t) + \sigma_t z$     ▷ Perform one denoising step
8: **end for**
9: **Return** $x_0$

---

Where $\alpha_t$ is the parameter that defines the noise schedule, i.e. the weighted mean between the data and noise at the noise level $t$, $\epsilon(x_t, t)$ is the noise level conditioned noise prediction network or alternatively the learned gradient of the data when considered from the score matching perspective, and $\sigma_t$ is the standard deviation of the noise at noise level $t$.

# 3 EXPERIMENTS

We first illustrate our method on a benchmark system, defined by Brownian dynamics of a particle in a 2-dimensional energy function. We prepared a training set by generating trajectory snippets with a physical simulator and trained a surrogate Diffusion Model to reproduce the original physical dynamics.

We then demonstrated the enhanced sampling capabilities of the model, as compared to the physical simulator, in the form of conditional generation, generation using constraints on the trajectory, and the generation of variations of trajectories. We then showcased our approach on a benchmark molecular system, namely Alanine Dipeptide.

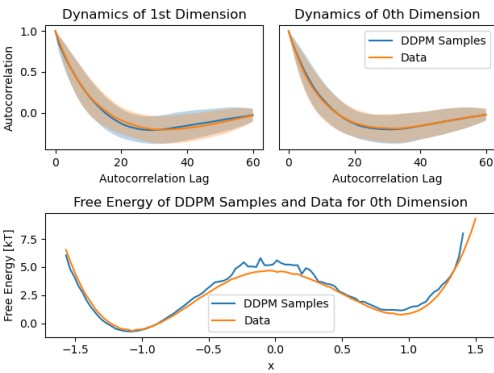

Figure 3: Free energy and dynamics comparison.

Lastly, we performed an ablation study to highlight the importance of the energy formulation, and the self-attention layers, and to study the quality and quantity of the training set produced by the physical simulators necessary to produce accurate results. The appendix section 'Experimental Settings' contains the experimental settings, including data generation and processing, model size, training hyperparameters, and an experiment pertaining to the physical validity of the enhanced sampling trajectories.

## 3.1 BROWNIAN MOTION IN A BENCHMARK POTENTIAL

To assess the reconstruction accuracy of DynamicsDiffusion and its ability to perform rare event sampling via inpainting, variation generation, and conditional generation, we generated trajectories by simulating the dynamics of a particle in a double well under Brownian motion. This simple system recapitulates a molecular rare transition between two metastable states. We sampled trajectories integrating Brownian motion with the Euler–Maruyama integrator on the energy $V(x, y) = -9x^2 + x^3 + 4.5x^4 + 3y^2$, setting the system's temperature at $k_\mathrm{B}T = 1$ for the unconditioned experiments, and in a range from $0.5$ to $1.5$ for the temperature-conditioned experiments.

After training the diffusion model, we systematically benchmarked it by comparing the thermodynamics and dynamics of the reconstructed trajectories with the ones obtained by the physical integrator. Trajectories sampled by the diffusion model qualitatively behave like physical trajectories. They mostly populate the bottoms of the energy wells, and only rarely cross the barrier in between 4. Additionally, trajectories sampled by conditioning at higher temperatures explore higher energy values, as one would expect given the physics of this system. We also quantitatively assessed the accuracy of the reconstruction by comparing the stationary probability distribution explored by the trajectories sampled from the diffusion model. In our case this distribution is proportional to the Boltzmann factor, $\propto e^{-\frac{V(R_\mu^{(i)})}{k_\mathrm{B}T}}$, which determines the thermodynamics of the system. Both original and reconstructed distributions match quantitatively across different temperatures.

The reconstructed dynamics were also accurate, as can be assessed by comparing the auto-correlation of training data trajectories and those generated by the model (figure 3).

We also illustrated the possibility of performing enhanced sampling of rare events by sampling a transition path crossing over the barrier performing inpainting, and variations of an existing transition path 4.

## 3.2 MOLECULAR SYSTEM

We performed a proof-of-concept application on the alanine dipeptide molecular system, which is a standard benchmark in the molecular simulations field. While the dynamics of the system occur in its full configuration space, i.e., the collections of three Cartesian coordinates for each atom in the molecule, a (relatively) rare conformational transition occurs around the dihedral angle $\psi$. For this

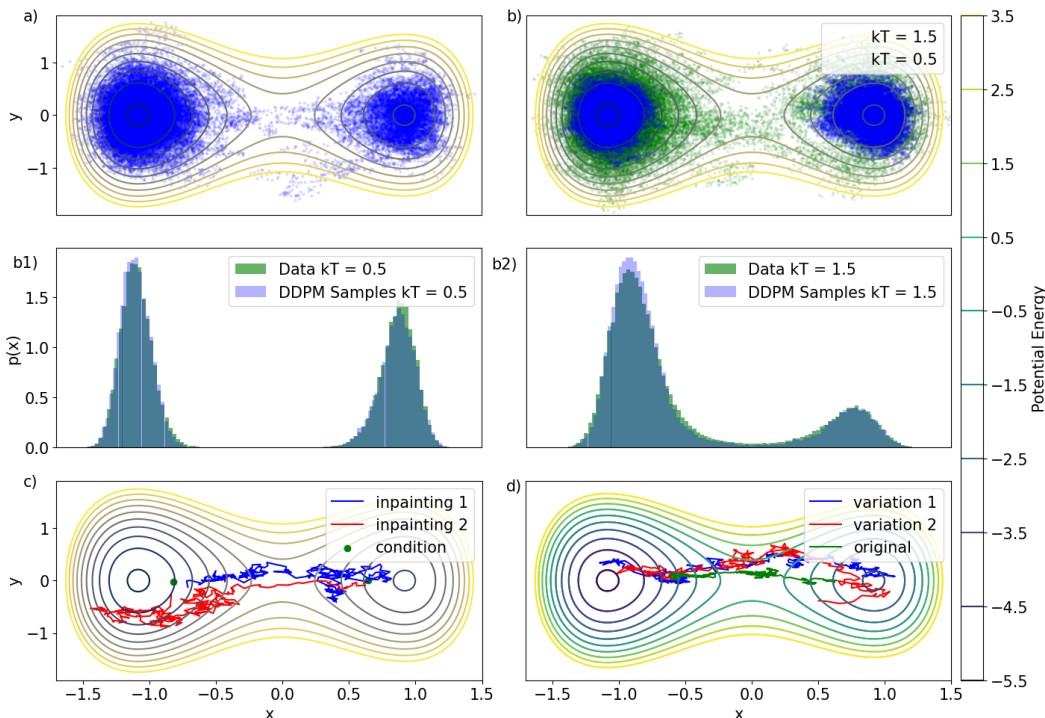

Figure 4: (Properties of the trajectory diffusion model) **(a) Sampling the Boltzmann Distribution**: The diffusion model is able to generate samples according to the Boltzmann distribution of the training data, including transitions. At inference, the sampling steps to generate a trajectory can be lowered from 1000 to 30 using numerical integration. **(b) Conditional generation**: The model can be trained in a conditional manner to generate samples that fulfill a condition, like a temperature or other physical properties. **(b1)** Comparison of the free energy of the model samples and the data at 0.5 $k_b T$. **(b2)** Comparison of the free energy of the model samples and the data at 1.5 $k_b T$.**(c) Inpainting/Spatial conditioned generation**: Another way to condition the generation of the model to perform enhanced sampling is to fix certain parts of the generated trajectories and let the model fill in the rest. This can, for instance, be used to sample state transitions by fixing the start and the end of the generated trajectory to the two states respectively, and letting the model generate the states between them. **(d) Trajectory variation generation**: The diffusion model can be used to generate similar trajectories or 'variations' of existing trajectories or even hand-crafted faux-trajectories by partially noising the trajectory and then letting the diffusion model denoise them again. The degree of noise added to the original trajectory determines the degree of similarity to the generated variations.

reason, it is common to represent the stationary distribution described by molecular trajectories on the low dimensional projection defined by two dihedral angles $\psi$ and $\phi$. Visualizing these angles in a Ramachandran plot of both the sampled and real trajectories allows one to validate the results of this 30-dimensional system visually.

We simulated the molecular trajectories for the training data using OpenMM with an amber99sb force-field saving every 10th frame leading to a time resolution of 0.02 picoseconds, since for a lower time resolution the performance of the model starts to degrade. The resulting Cartesian atomic coordinates were filtered for hydrogen atoms and aligned to a reference frame to remove rotation and translation symmetries from the data.

We computed the free energy of the training data and the samples of the DDPM in the coordinate system by the two dihedral angles 5. We evaluated the accuracy of the reconstructed dynamics by comparing the autocorrelation of the trajectories mapped onto these two angles. The results, seen in 5, show that the model can capture the dynamics and the free energy of the system. The model even sampled a rare transition along the $\Psi$ angle, which was not observed in the simulation that generated the training data but is theoretically possible, pointing to the generalization capabilities of the method.

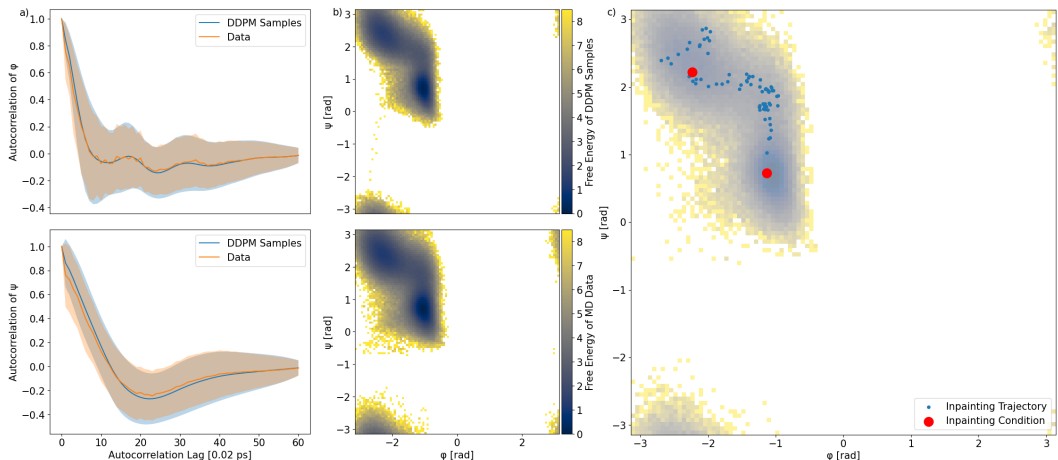

Figure 5: (Properties of the trajectory diffusion model) **(a) Dynamics Comparison**: The dynamics of the DDPM samples and the simulated samples were compared via an autocorrelation, $\pm$ its standard deviation, along a time lag. **(b) Free Energy Profile**: The free energy of both the mode, top, and the simulated samples, bottom. **(c) Inpainting/Spatial conditioned generation**: Example trajectory between two states of Alanine Dipeptide generated via inpainting.

## 3.3 ABLATION STUDIES

We performed ablations to see the effect of the design choices we made and to identify crucial components. To evaluate the performance of the models, we trained on the double-well potential and evaluated the free energy mismatch between samples and the ground truth along the x-axis, which was chosen as it is the reaction coordinate of the system. This mismatch was computed for 100 bins of the x-coordinate and averaged. Ablations were also performed on the size and trajectory length needed to train a good *DynamicsDiffusion* model, which can be found in the appendix.

**Energy-model formulations versus force formulation**  To assess the inductive bias we introduce. We train conventional UNets and UNets with the additional energy inductive bias N=10 times per model, for which the mean and standard deviation were computed. The results in Table 9 show that the energy formulation outperforms the force formulation of the UNet.

**Self-Attention versus fully convolutional**  To examine the effect of the self-attention layers the same experimental setup as above was used. Without one self-attention layer per block, the UNet is a fully convolutional model. The results can be found in Table 9.

Table 1: Energy vs force and self-attention vs attention-free model comparison

| Formulation | Free Energy Error $k_b T$ | Free Energy Error Transition-Region $k_b T$ |
|---|---|---|
| Energy | 0.36±0.19 | 0.63±0.36 |
| Force | 0.55±0.32 | 0.86±0.67 |
| Self-Attention & Energy | 0.17±0.19 | 0.33±0.19 |
| No-Attention & Energy | 0.34±0.34 | 0.30±0.33 |

## 4 CONCLUSIONS

We proposed the first deep generative model to generate accurate trajectories of a molecular system which can serve as a useful surrogate model of the physics-based simulator, due to its enhanced sampling capabilities. This work opens up many new avenues of research. One limitation of the current model is that it does not generalize well beyond a certain generation time trajectory length and

can only accurately generate trajectories for time horizons in the ranges it has seen during training. Another exciting avenue of improvement is to generalize the method to work for any molecular system, not just the one it has been trained on, by changing the underlying diffusion model to a GNN, and thereby due to the parameter-sharing nature of the system making it system size agnostic. Another avenue that can also be approached using GNNs is to include the symmetries of the system better by using new GNN architectures that are equivariant to translations, rotations, and permutations like those proposed by Satorras et al.. Adopting the newly proposed consistency network Song et al. (a) training method could also give the method more flexibility when choosing between generation speed and quality. Further research can go into scaling this method to larger systems, like larger proteins whose dynamics and state transitions are of scientific interest.

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

# A  APPENDIX

## THE NOISING & DE-NOSING PROCESS

In this section, we describe the noising process of the denoising diffusion probabilistic model (DDPM), which is utilized to generate noisy observations from the original trajectory snippets in the training data $\mathcal{D}$. The noising process, also called the forward diffusion process, consists of a series of steps that introduce noise to the samples till the samples are fully noised. The model is then trained to invert this process to generate samples $x_i$ from noise.

A DDPM Sohl-Dickstein et al. comprises two Markov chains: a forward chain that perturbs data, and in our case trajectory snippets, to noise and a reverse chain that restores the noise to samples from the same distribution as the training data. In our case, the final trained model can then sample trajectories. The forward process transforms the complex data distribution into a simple prior distribution in the form of a standard normal distribution. The reverse chain, parameterized by deep neural networks, learns to reverse the process. New trajectory snippets $x_i$ can then be generated by sampling a random vector from the prior distribution and employing ancestral sampling through the reverse Markov chain.

The forward process can be described both using an iterative noise addition or a one-step addition that jumps straight to a noise level $t$. They can be described as follows: For a distribution $x_0 \sim q(x_0)$, equivalent to the data distribution $x \sim q(x)$ with no added noise, the forward process generates a sequence of random variables $x_1, x_2, \ldots, x_T$ given by a transition kernel $q(x_t|x_{t-1})$. The transition kernel is designed to turn a complex data distribution into a simple prior distribution, usually the Gaussian distribution, via a Gaussian perturbation:

$$q(x_t|x_{t-1}) = \mathcal{N}(x_t; \sqrt{1-\beta_t}x_{t-1}, \beta_t I) \qquad (4)$$

here $\beta_t \in (0, 1)$ also known as the beta schedule is chosen to determine how much noise is added at each step of the forward process. Different schedules, like linear, scaled linear, or cosine Karras

et al., and their strength should be picked based on the training data to ensure an even loss of data information along the forward process noising trajectory.

The formula for going directly to noise level $t$ from level $0$ without the intermediate steps can be attained by using the Markovian properties of the process, see Sohl-Dickstein et al.

$$q(x_t|x_0) = \mathcal{N}(x_t; \sqrt{\alpha_t}x_0, (1 - \alpha_t)I). \tag{5}$$

Where $\alpha_t = 1 - \beta_t$. With this relationship, we can go from a sample to level t by sampling a noise vector $\xi \sim \mathcal{N}(0, I)$ of the same shape as the data vector, in our case $\mathbf{A} \in \mathbb{R}^{b \times \tau \times N \times 3}$, and is then mixed with the sample scaled by $\alpha_t$:

$$x_t = \sqrt{\alpha_t}x_0 + \sqrt{1 - \alpha_t}\xi. \tag{6}$$

Conceptually, the forward process incrementally adds noise to the trajectory snippets until all structural information is lost.

To generate new trajectory snippets, DDPMs first generate an unstructured noise vector from the prior distribution (which is generally simple to obtain) and then progressively remove the noise by applying a learnable Markov chain in the reverse time direction. Specifically, the reverse Markov chain is parameterized by a prior distribution $p(x_T) = \mathcal{N}(x_T; 0, I)$, identical to the one the forward process ends up with, and a learnable transition kernel $p_\theta(x_{t-1}|x_t)$ parameterized by a neural network with weights $\theta$ and take the form of:

$$p_\theta(\mathbf{x}_{t-1} \mid \mathbf{x}_t) = \mathcal{N}(\mathbf{x}_{t-1}; \mu_\theta(\mathbf{x}_t, t), \Sigma_\theta(\mathbf{x}_t, t)) \tag{7}$$

Where the mean $\mu_\theta(\mathbf{x}_t, t)$ and variance $\Sigma_\theta(\mathbf{x}_t, t)$ are extracted from the outputs of the network, although in certain DDPM formulations, the variance is not learned. The parameters of the model are optimized using the following loss function:

$$Loss = \mathbb{E}_{t \sim \mathcal{U}[1,T], \mathbf{x}_0 \sim q(\mathbf{x}_0), \epsilon \sim \mathcal{N}(\mathbf{0}, \mathbf{I})} \|\epsilon - \epsilon_\theta(\mathbf{x}_t, t)\|^2 \tag{8}$$

Here the noise vector $\epsilon_\theta$ that the model predicts is compared to the ground truth $\epsilon$. The model makes this prediction based on the input of a sample $\mathbf{x}_0$ noised to $\mathbf{x}_t$ using the forward noising process and a noise level $t$ sampled uniformly $\mathcal{U}(1, T)$.

## MOLECULAR DYNAMIC TRAJECTORIES AS A DATATYPE

The training data is denoted as $\mathcal{D} = {x_i}{i = 1}^K$, representing trajectory snippets from a probability density $p(x)$. Each $x_i$ is a trajectory of $\tau$ time steps, with $\tau$ uniformly sampled from $[\tau min, \tau_{max}]$. A trajectory snippet, $x_i$, is an ordered set of system configurations over time, $x_i = {R^{(i)}\mu}{\mu = 1}^\tau$. For a system of $N$ atoms, a configuration $R_\mu$ is the set of Cartesian coordinates for each atom, $R_\mu \in \mathbb{R}^{N \times 3}$.

Training data is batched into arrays of shape $(b, \tau, N \times 3)$, where $b$ is the batch size. A batch, $\mathcal{B}$, is denoted as $\mathcal{B} = {\mathcal{D}_k}{k = 1}^b$. Each trajectory snippet in a batch, $\mathbf{x}_i^{(k)}$, has $\tau$ time steps, each with $N$ atoms and their 3D coordinates. The full data array for a batch is $A \in \mathbb{R}^{b \times \tau \times N \times 3}$. This data can be likened to an image in diffusion models, with the time dimension analogous to image width. Refer to the accompanying figure for a visual representation of the data shape.

Data Shape of the Trajectory Snipets

Figure 6: Illustration of one batch entry, a trajectory snippet, with its time and atomic coordinate dimension.

EXPERIMENTAL SETTINGS

DOUBLE WELL EXPERIMENTS

BROWNIAN MOTION IN A DOUBLE WELL

In this section, we provide a detailed description of the numerical integration of Brownian motion in a double well for generating training data for basic experiments and conditional generation. The key components of our simulation setup are summarized in Table 2.

Table 2: Summary of Brownian motion in a double well

| Parameter | Value |
|---|---|
| Dynamics | Brownian |
| Integration Method | Euler–Maruyama method |
| Time Step | 0.001 |
| Total Simulation Time | $2.3 * 10^9$ |
| Potential | $-9x^2 + x^3 + 4.5x^4 + 3y^2$ |
| Mass | 1 |
| $k_{\mathrm{B}}T$ | 1 |
| Conditional Generation Specifics | |
| $k_{\mathrm{B}}T$ Range | [0.5, 1.5] |

**Data Processing** The raw data obtained from the simulations were processed before being used as input for the deep learning model. The data was re-scaled to the range of [-1, 1] for both dimensions. For the training of the temperature-conditioned model, the $k_{\mathrm{B}}T$ values used for conditioning were also rescaled to the range of [-1, 1].

ARCHITECTURE AND TRAINING HYPERPARAMETERS

In this section, we provide a detailed description of the experimental settings used during the training run of our deep learning model on the double well data. The key components of our experimental setup are summarized in Table 6.

Table 3: Summary of experimental settings

| Parameter | Value |
|---|---|
| Optimizer | Adam |
| Batch Size | 512 |
| Learning Rate | 1e-4 |
| Learning Rate Finetune | 1e-6 |
| Batch Count | 3000 |
| Batch Count Finetune | 1000 |
| Snippet length $\tau$ | 64 |
| Number of Parameters | 646337 |
| UNet Block Multiplier | [1, 2] |
| Input Embedding Channels | 64 |
| Activation Function | Mish |
| EMA Decay | 0.9975 |
| EMA Update Frequency | 10 |

OPTIMIZER AND HYPERPARAMETERS

We employed the Adam optimizer with a learning rate of 1e-4 for the first 3000 batches and then 1e-6 for the last 1000 batches. Batches were sampled by taking random snippets from the training trajectory of length $\tau = 64$. The batch size was set to 512. A moving average of the training model weights were used to evaluate the model during training and post-training inference, this EMA model was updated every 10 batches with a decay of 0.9975.

MODEL ARCHITECTURE

Our UNet consists of 646337 parameters. The UNet comprises multiple blocks with a channel count multiplier of [1, 2]. Each block in the encoder part of the UNet consists of two temporal ResNet blocks, consisting of 1D convolutions, a residual connection across the entire block, and a down-sampling operation, also in the form of a 1D convolution. The Encoder is followed by a middle block consisting of two temporal ResNet blocks and a residual connection. The Decoder part of the UNet is symmetric to the Encoder except for the swap of the down-sampling operation to an up-sampling one, consisting of a transposed convolution.

ENCODER-ONLY ARCHITECTURE EXPLORATION

Due to the nature of the Energy formulation, a decoder is technically not necessary since even with only an encoder that outputs a singular value per trajectory, i.e. a 'trajectory energy' the Jacobian computation recovers a tensor equivalent in shape to the model input. To test such a model's performance, we ablated the decoder of the UNet, and therefore the encoder-decoder residual connections too, and employed the same energy head as in the UNet approach. The encoder-only model has fewer parameters than the full UNet, hence we performed a second benchmark with a slightly larger encoder to compensate for the lower parameter count.

RESULTS

Table 4: Full U-Net vs. encoder only model comparison

| Formulation Transition-Region $k_b T$ | Parameter Count | Free Energy Error $k_b T$ | Free Energy Error |
|---|---|---|---|
| Attention & Energy | 843k | 0.17±0.19 | 0.33±0.19 |
| Encoder only & Attention & Energy | 513k | 0.29±0.16 | 0.77±0.20 |
| Encoder only & Attention & Energy | 900k | 0.19±0.15 | 0.60±0.12 |

The results show that the encoder-only model performs worse than the full UNet, even when accounting for the difference in parameter count, however, the performance gap is not extensive, making it worth pursuing in future research.

## ALANINE DIPEPTIDE EXPERIMENTS

### OPENMM SIMULATION PARAMETERS

In this section, we provide a detailed description of the OpenMM simulation parameters used for generating the training trajectories for the Alanine Dipeptide system in a Vacuum. The key components of our simulation setup are summarized in Table 5.

Table 5: Summary of OpenMM simulation parameters

| Parameter | Value |
|---|---|
| Force Field | amber99sb |
| Integration Method | Langevin |
| Time Step | 0.002 ps |
| Temperature | 320 K |
| Friction Coefficient | 1 1/ps |
| Simulation Time | 20 ns |
| Saving Interval | 0.02 ps |
| nonbondedCutoff | 1 nm |
| constraints | HBonds |
| nonbondedMethod | NoCutoff |

**Data Processing**   The raw data obtained from the OpenMM simulations underwent several processing steps before being used as input for the deep learning model:

1. **Hydrogen Removal**: All Hydrogen atoms were removed from the trajectory.

2. **Trajectory Alignment**: All frames were aligned along the conformation of a reference frame.

3. **Data Scaling**: The atomic coordinates were re-scaled to a range of [-1, 1].

### ARCHITECTURE AND TRAINING HYPERPARAMETERS

In this section, we provide a detailed description of the experimental settings used during the training run of our deep learning model on the Alanine Dipeptide data. The key components of our experimental setup are summarized in Table 6. The UNet was trained for three different sizes to find the smallest model that was still able to reconstruct the free energy landscape for the benchmark, however, the main results are based on the largest model. The different model configurations are comma-separated in Table 6.

Table 6: Summary of experimental settings

| Parameter | Value |
|---|---|
| Optimizer | Adam |
| Batch Size | 256 |
| Learning Rate | 4e-5 |
| Learning Rate Finetune | 1e-6 |
| Batch Count | 30000 |
| Batch Count Finetune | 10000 |
| Snippet length $\tau$ | 64 |
| Number of Parameters | 10201729, 2583937, 1458721 |
| UNet Block Multiplier | [1, 2, 4], [1, 2], [1, 2] |
| Input Embedding Channels | 128, 128, 96 |
| Activation Function | Mish |
| EMA Decay | 0.9975 |
| EMA Update Frequency | 10 |

## OPTIMIZER AND HYPERPARAMETERS

We employed the Adam optimizer with a learning rate of 4e-5 for the first 30,000 batches and then 1e-6 for the last 10,000 batches. Batches were sampled by taking random snippets from the training trajectory of length $\tau = 64$. The batch size was set to 256. A moving average of the training model weights was used to evaluate the model during training and post-training inference, this EMA model was updated every 10 batches with a decay of 0.9975.

## MODEL ARCHITECTURE

Our UNet consists of 646337 parameters. The UNet comprises multiple blocks with a channel count multiplier of [1, 2, 4]. Each block in the encoder part of the UNet consists of two temporal ResNet blocks, consisting of 1D convolutions, a residual connection across the entire block, and a down-sampling operation, also in the form of a 1D convolution. The Encoder is followed by a middle block consisting of two temporal ResNet blocks and a residual connection. The Decoder part of the UNet is symmetric to the Encoder except for the swap of the down-sampling operation to an up-sampling one, consisting of a transposed convolution.

## DATA ABLATION EXPERIMENTS

In this section, we provide a brief description of the data ablation experiment. The experiment evaluates the model's performance when trained on datasets of different sizes and trajectory lengths. The data size is decreased by a factor of 10 four times, and we measure the model's performance on both full transition (long trajectory) and no full transition (short trajectory) datasets. The long trajectory dataset consists of 6900 trajectories where each trajectory is 8000 steps long while the short trajectory dataset consists of 883200 trajectories where each trajectory is 64 steps long. The overall training data size, when measured in steps, is nearly identical. The aim of the experiment is to ascertain the size and shape of data that the method needs. The choice of the two different dataset shapes, a few long vs many short trajectories, are there to see if the model is able to learn from trajectories that are small and individually do not show relevant events, like a full transition. Given that small dataset, subsamples do not accurately follow the distribution of the system in equilibrium the mismatch between the data subset and the equilibrium free energy is also computed as a reference. The performance is evaluated in the same manner as in section 3.1 where the free energy is computed along the x-direction of both samples taken from the model and the data distribution of the full data.

## RESULTS

The following table shows the results of the data ablation experiment. The first column represents the dataset size, the second shows the model's performance on the short trajectory dataset, the third column displays the model's performance on the long trajectory dataset, and the last two columns

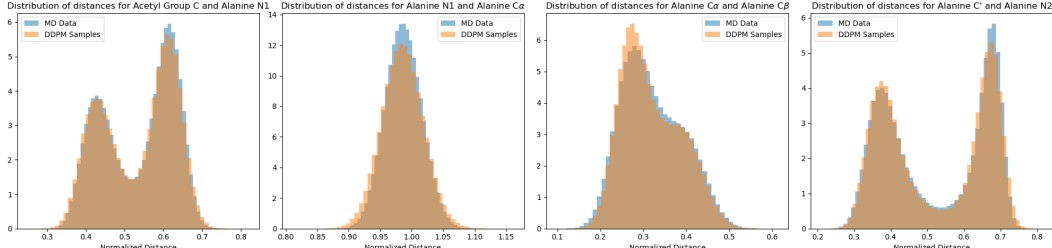

Figure 7: Pairwise distances between atoms in the generated Alanine Dipeptide trajectories and the ground truth trajectories.

show the mismatch between the true underlying data distribution and the training samples for the short and long trajectory data subsamples respectively.

Table 7: Model performance after training on datasets of different sizes and trajectory lengths.

| Data Size | Error Short Traj. | Error Long Traj. | Data Error Short | Data Error Long |
|---|---|---|---|---|
| 1/1 | $0.25 \pm 0.13$ | $0.32 \pm 0.12$ | $0.0 \pm 0.0$ | $0.1 \pm 0.0$ |
| 1/10 | $0.28 \pm 0.29$ | $0.27 \pm 0.14$ | $0.01 \pm 0.0$ | $0.03 \pm 0.01$ |
| 1/100 | $0.24 \pm 0.11$ | $0.39 \pm 0.28$ | $0.01 \pm 0.01$ | $0.06 \pm 0.04$ |
| 1/1000 | $0.22 \pm 0.17$ | $0.17 \pm 0.14$ | $0.02 \pm 0.02$ | $0.30 \pm 0.23$ |
| 1/10000 | $0.32 \pm 0.22$ | $0.64 \pm 0.49$ | $0.09 \pm 0.07$ | $1.16 \pm 0.91$ |

Based on the results presented in Table 7, we observe that training on either many short or a few long trajectories does not make a difference up until the data is very sparse, where the training data consisting of short trajectories pulls ahead. However, in this case, it is likely due to the fact that the short trajectories are more representative of the distribution of the underlying system as the trajectory starts are picked from equilibrium trajectories, as compared to the singular long trajectory remaining when only $1/10000$ data is considered and thus the likelihood of this trajectory not being representative is very high, as seen by the much higher mismatch between the $1/10000$ subsample and the full data in the long trajectory scenario. Another takeaway from the data ablations is, that for the low data regime, the mismatch between the model and the underlying system is smaller than the mismatch between the data subsample and the underlying system, which hints at a strong inductive bias in the model, helping it to learn to generate trajectories more accurately than the non-representative training data.

### ALANINE DIPEPTIDE BOND LENGTH DISTRIBUTION EVALUATION

We calculated bond length distributions for trajectories generated by the model on Alanine Dipeptide to further validate its accuracy in capturing system dynamics by comparing a few selected bond lengths to the ground truth.

### RESULTS

### ALANINE DIPEPTIDE DATA ABLATION

Due to the difficulty of obtaining molecular dynamics trajectories, we performed a triangulation of the minimal data needed to train a DynamicsDiffusion model to accurately generate Alanine Dipeptide trajectories. We find that for Alanine Dipeptide 200 picoseconds of training data are sufficient as long as they cover all relevant states even for generalization toward the state borders.

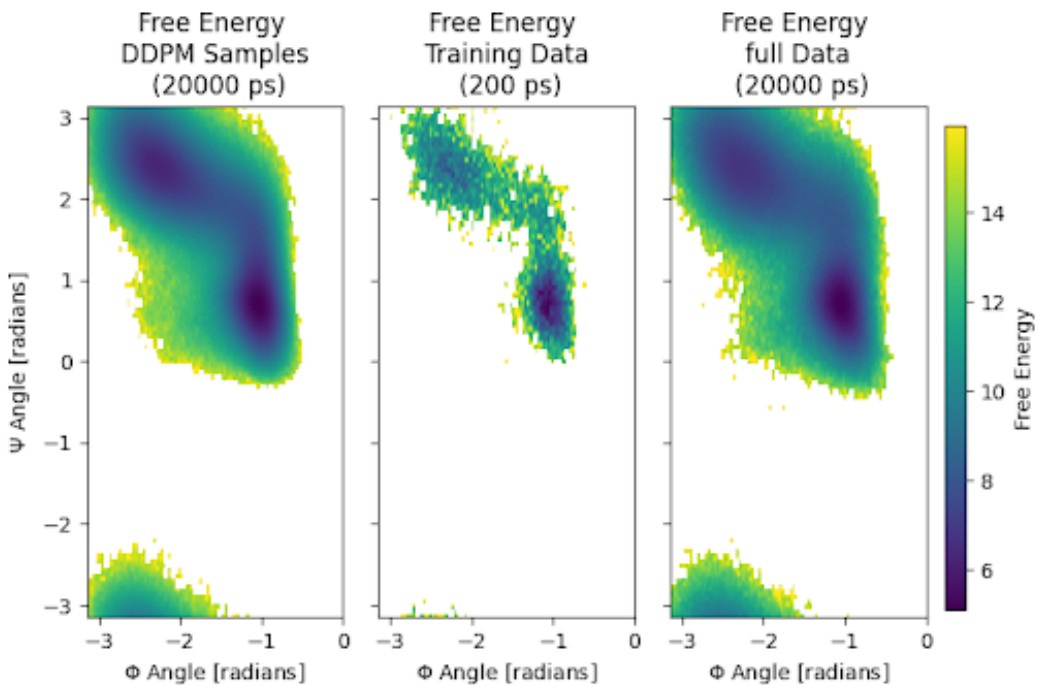

Figure 8: Alanine Dipeptide trajectories projected onto their dihedral angles. In order are the DDPM samples, the training data for the data ablation experiment, and the full simulated data.

RESULTS

GENERATION STEP COUNT EXPERIMENTS

Discrete Diffusion Models are trained with a fixed count of steps along the denoising trajectory, usually 1000, however, this means, that naively to generate samples the UNet of the Diffusion model needs to be called 1000 times, which leads to extremely slow sampling. However since the denoising process, like the noising process, is a differential equation it can be numerically integrated. To that end, people have used a variety of numerical integrators to generate samples from the trained diffusion model Karras et al.. With this, the step count can be brought down to a fraction of the original training steps. However, as with conventional numerical integration a trade-off between accuracy and speed exists. To ascertain how this trade-off affects our model we computed the free energies of the double well model at the full 1000 steps and then compared the free energy to samples generated using a discrete Euler sampler at lower step counts.

RESULTS

The results seen in **??** seem to indicate, that at around 50 steps the error seems to increase but that the mismatch between the free energy profiles only increases sharply below 30 steps, implying a range from 30 to 50 steps is suitable depending on one's accuracy needs.

VALIDATING THE PHYSICAL CORRECTNESS OF INPAINTING

To validate the physical plausibility of the enhanced sampled transition trajectories the mean of the mean squared displacement, MSD, and its standard deviation were computed across the transition region for the sampled trajectories (Inpainting and Variation) and those generated with the numerical simulator as the ground truth. For this trajectories were generated via inpainting from one state to another by inpainting frames at x = -0.89, 1.02, then all trajectories crossing from one x-Coordinate value to another, defined by boundaries at -0.63 and 0.65 respectively, were filtered and the MSD was computed across that region. One failure case of inpainting is the creation of non-continuous

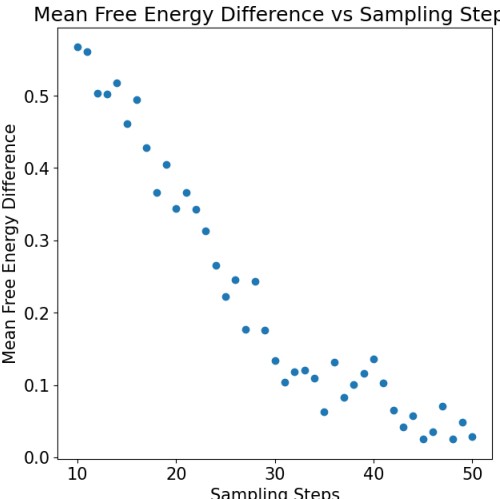

Figure 9: The sampling steps of the discrete Euler scheduler vs. the error of the samples as compared to those generated with 1000 in mean absolute free energy along the x-axis.

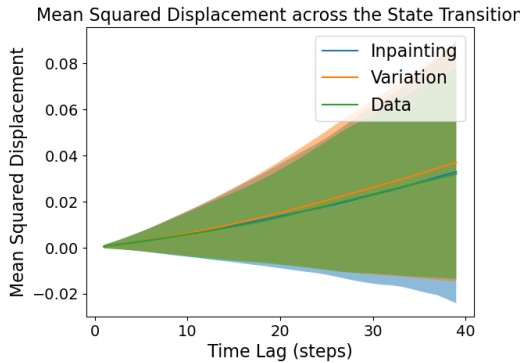

Figure 10: MSD of the enhanced sampling data across the transition region as compared with ground truth distribution.

trajectories which can occur. The frequency of this depends on the choice of inpainting hyperparameters, namely the inpainted trajectory time relative to the distance the trajectory has to cover and the number of frames imposed. To filter our noncontinuous trajectories we filtered out all trajectories that have too few time steps present in the transition region. Enhanced sampling trajectories via variation generation were generated by taking a simulated transition trajectory and then generating variations of this trajectory by nosing to $t = 0.55$ and then denoising them, afterward the same transition region selection was applied as in the ground truth data and the inpainting enhanced sampling.

MULTI-LENGTH GENERATION EXPERIMENTS

We wanted to assess how well the model could produce trajectories of varying lengths, based on its training data. To do this, we trained the DynamicsDiffusion model using trajectory lengths ranging from 32 to 326 time steps. It's important to note that these lengths correspond to 320 to 3260 time steps in the simulator, as the model operates at a time resolution 10 times lower. After training, we sampled the model at both the shortest and longest lengths from this range. We then compared the distribution of states in these samples to the actual states from the simulator.

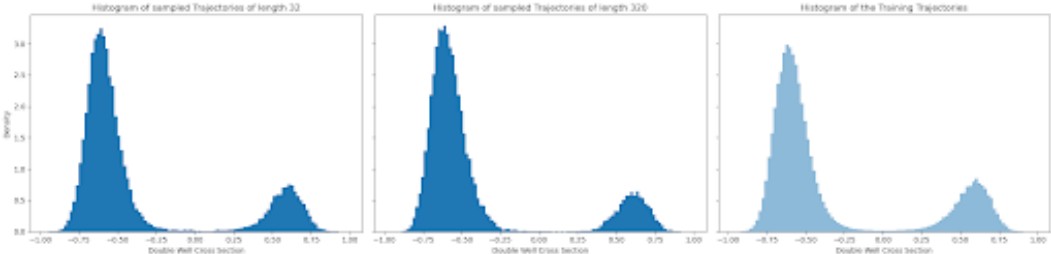

Figure 11: DynamicsDiffusion trajectory distribution for the Double Well systems sampled at different trajectory lengths.

Table 8: Benchmarks of differently (1.4 million to 10.2 million parameters) sized DDPMs for Alanine Dipeptide (AD) trajectory generation

| Simulator | Gromacs | AD 10.2M | AD 2.6M | AD 1.4M |
|---|---|---|---|---|
| Speed [ns/day] | 6711.15 | 9276 | 14921 | 15639 |

RESULTS

BENCHMARKS

The speed of DynamicsDiffusion was compared to Gromacs Abraham et al., a commonly used molecular dynamics simulation software, see Table 8, for the generation of Alanine Dipeptide. The setup of the simulation and the model inference can be found in the appendix, along with a section where the optimal number of denoising steps is determined for the numerical integration of the denoising process Karras et al.. Furthermore, it is worth noting that this benchmark does not include model training and only inference and was only performed in one hardware configuration.

A.1 JACOBIAN COMPUTATION MODEL SLOWDOWN ALANINE DIPEPTIDE MODEL

To assess the incurred slowdown of the introduction of the Jacobian computation necessary for the energy formulation we benchmarked the model with and without the energy head and compared their relative forward pass speeds.

Table 9: Jacobian Computation Slowdown

| Formulation | Parameter Count | Relative Slowdown Factor to Baseline |
|---|---|---|
| No Energy Formulation (Baseline) | 6.438M | 1 |
| Energy Formulation | 6.438M | 2.46 |

SOFTWARE AND HARDWARE

- Programming Language: Python 3.9
- Deep Learning Library: PyTorch 1.9
- GPU: NVIDIA A6000

