# OpenReview forum: "DynamicsDiffusion: Generating and Rare Event Sampling of Molecular Dynamic Trajectories Using Diffusion Models"
_ICLR.cc/2024/Conference — Submitted to ICLR 2024_

### Official Review · Reviewer_3FPR · 2023-10-13

**Soundness:** 2 fair
**Presentation:** 3 good
**Contribution:** 2 fair
**Rating:** 3
**Confidence:** 5

**Summary:**

This paper considers the rare event sampling problem in the molecular dynamics simulation which aims to sample the transition state (with a high energy barrier) between two metastable states. Specifically, this paper proposes to solve this problem with a diffusion model that learns to simulate such trajectories (transition paths between the two metastable states). The authors leverage the inpainting flexibility of diffusion models to inpaint missing dynamics in the trajectories. Experiments are conducted on both a synthetic and a real-world molecular dynamics simulation problem.

**Strengths:**

* Sampling rare events and transition paths is a crucial problem in computational chemistry and has wide applications in studying a wide range of chemical reactions such as protein folding, protein-ligand binding, etc.
* The proposed method that uses the diffusion model to model the distributions of transition paths and sampling new transition path is technically sound.

**Weaknesses:**

* The major concern of this work is both the generalizability and scalability of the proposed method. From the method description and experiment section, this method requires sampling a path (though seems to be short, 10 timesteps) as the training data. There is no evidence that demonstrates this could generalize to longer-time dynamics or the transition path (which even requires crossing the energy barrier) nor scale to larger systems since the size of the system scales with the number of the frames used (and I believe for more complicated systems, more frames are needed).
* It seems the different timestep of the path need to be properly isolated when running the diffusion models, otherwise they may affect each other.
* This is not a major concern but the proposed method itself does not contribute to the technical novelty of this paper as it mostly uses diffusion models developed in previous work and deployed in this problem.
* I am also worried about how physical the generated paths would be. Even though the authors demonstrate the distribution plots and how the dihedral angels change, from my personal experience, sometimes the paths could completely fail while the dihedrals still look good. It would be good to see more evidence in terms of how physical the generated paths are or visualizing some paths.
* Some related work in this area is missing. [1] is also a directed generation approach to find the transition path between two metastable states.

[1] Holdijk, L., Du, Y., Hooft, F., Jaini, P., Ensing, B. and Welling, M., 2022. PIPS: Path Integral Stochastic Optimal Control for Path Sampling in Molecular Dynamics.

**Questions:**

See weaknesses.

---

### Official Review · Reviewer_2a4b · 2023-10-24

**Soundness:** 2 fair
**Presentation:** 1 poor
**Contribution:** 2 fair
**Rating:** 3
**Confidence:** 5

**Summary:**

In this work the authors present an extension of the standard application of diffusion models in the context of molecular systems to additionally study the dynamics of the systems. The authors achieve this by generating trajectories trained on MD simulation data instead of single instances of the system. The proposed method is evaluated using both a 2D toy example as well as small molecular system in the form of Alanine Dipeptide.

**Strengths:**

**Originality:** The method lacks algorithmic novelty, most of the presented work are adapted from prior work on diffusion models for time series and molecular data, but this is alleviated by the originality the paper has in applying these previously established techniques to a novel application area. Using Diffusion models for generating molecular transitions is in itself an interesting and novel idea.

**Significance:** The paper addresses and important problem in (computational) chemistry, the sampling of rare events.

**Weaknesses:**

Unfortunately the paper lacks in quality and clarity. I've elaborated on this below in my detailed comments and requests for clarifications, but in general the papers notation is hard to follow and unfortunately the experiments do not address the claims made.

**Questions:**

Please find my detailed list of comments, ordered by section, below.

Where needed, I indicated that something might be a breaking issue. These issue will have to be either clarified or rectified before I will consider increasing my vote to accept.

**Introduction:**
1. This also pertains to the remainder of the paper, please make sure the citations are correctly placed between brackets unless when used as part of the narrative.
2. The first paragraph of the introduction state "However, their potential in the study of the dynamics of molecular system is still unexplored" when discussing the use of Diffusion models for generating ensembles of molecular conformations. This sentiment is reflected a number of more times throughout the introduction and the remainder of the paper, as such, I would expect the experiments to focus on properties related to the dynamics of the system that can not be determined from the ensembles provided by standard Diffusion models. However, given that the experimental section mostly focusses on evaluating Free Energy estimates (which can be derived from the equilibrium distribution), this does not seem to be the case. I would like the authors view on this observation.
3. Similarly, this paragraph also states "an accurate description is essential to understand the biological functions of proteins and other molecules.". I urge the authors to make these "biological functions" concrete. Are they talking about binding affinity, transition states or some other properties? This distinction is important to be able to judge the contribution of the proposed method.
4. Related to my prior statement, the last sentence of the first paragraph states "otherwise rare but important events". I would appreciate a clear statement as to why these events are important and would consequently expect the experimental evaluation to focus on correctly identifying them.
5. The authors state that "standard classical atomistic MD must use an integration time step of 2fs". While this is an often used value, there is no specific requirement for MD to use a 2fs timestep. The word "must" should be reconsidered.
6. Paragraph 3, 4, and 5 discuss different approaches for sampling rare events, but somewhat lacks organisation. This is largely due to not mentioning general themes or creating a clear topology of the different methods. My understanding is that paragraph 3 discusses ML based methods for reducing the computational cost of determining the molecular forces, paragraph 4 discusses enhanced sampling methods for enforcing transitions, and paragraph 5 discusses direct sampling from the equilibrium distribution using ML frameworks. These overarching themes should be made more clear. In general, I think the paper would benefit from a dedicated related work section that goes more in depth for these 3 paragraphs. The introduction would in that case only need to focus on the most important methods.
7. Paragraph 3 currently has to narrow of a focus when discussing neural force fields and coarse graining. These are large research fields and can't be done justice with only specific citations. For example, neural force fields come in many other flavours then just the one based on GNNs. To resolve this issue, I would suggest proving citations to survey papers. For example, for machine learning force field [1] is well suited, and for coarse graining [2] provides a clear overview.
8. For paragraph 4, the discussion of large energy barriers should be part of the earlier discussion on timescales. It should be made clear that "rare events" does not refer to events that happen over a long period in time, but instead to events that happen with very low probability.
9. For paragraph 4, based on the authors statement to focus on system dynamics, TPS is in my opinion an extremely related topic and requires some additional discussion. For example, an additional discussion of ML based alternatives/extensions here would be appropriate [3, 4, 5, 6]
10. For paragraph 6, the authors use the term "enhanced sampling". From my understanding, this term is often used for methods that augment MD simulations (often by introducing a bias potential or other driving force). This does not seem to be the case for the suggested method.
11. The list of capabilities of the proposed method states "Generate trajectories that are conditioned to a global parameter like temperature". Based on my own understanding and the presented results, I am not confident that other global parameters, for example friction, will be possible.
12. Similarly the list of capabilities of the proposed method states "Generate an ensemble of (reactive) trajectories by partially noising and denoising them". I did not find any experimental validation of this. Additionally, a statement such as this would need to be validated to show that the sampling converges to the right distribution over trajectories.

**Diffusion Generative Modelling for Trajectories**
1. On first reading it is very hard to understand the description of the dataset. And after reading it a couple of times, it is still unclear to me if "trajectory snippet" refers to a single molecular configuration or a short sequence of configurations. Based on the first paragraph, I would assume that the dataset contains trajectories. $p(x)$ would thus be the distribution over trajectories of the system. Ie. $x$ here refers to a trajectory of arbitrary length? If so, it would be beneficial if the authors could include a more formal definition of this probability in terms of the marginal distirbution $p(R_0)$ and the transition distribution $p(R_i | R_{i-1})$. However, based on the second paragraph, i get the impression that $x_i$ is a single configuration, and instead $\boldsymbol{x}_i$ is a short trajectory. This confusion is a breaking issue for me and should be clarified by the authors.
2. The third paragraph, discussing the complete data array is hard to understand. Do the authors intend to clarify in this paragraph that various length of trajectories can be used?
3. In equation one, $\nabla_x$ and $p(x)$ should respectively be $\nabla_{x_i}$ and $p(x_i)$. To prevent further confusion with indexing, I would also suggest to use t instead of i for the updates.
4. Below the equation the authors state "$\epsilon\rightarrow 0$  we sample one $x_k$ from $p(x)$". I'm unsure what this statement refers to. If $\epsilon\rightarrow 0$ we have no update and thus the sample remains constant. This is independent on whether or not $K\rightarrow \infty$.
5. Pertaining the same sentence, and related to my earlier point, is $p(x)$ here the distribution over samples, or the distribution over trajectories?
6. Regarding the last sentence of this same paragraph, the authors refer to a section in the appendix by name here. This should replaced by a forward reference. This happens on a number of occasions throughout the paper.
7. For equation 2, the notation $R^{(i)}_{\nu, t=0}$  is unclear. Instead of adding the $t=0$ in the subscript would $R^{(i)}_{0}$ not suffice?
8. I'm unable to follow the authors discussion of the conservative forces and the role the modelling of the forces as the gradient of an energy function plays in this. Especially the claim that UNets do not satsify this property should be substantiated.
9. Regarding the last sentence of this paragraph, the result presented in the ablation study shows that this difference is not significant.
10. My understanding is that the authors suggest to use a model architecture that predicts an energy independently for each sample, then sums thism and then takes the gradient to obtain the final force, which is used as the predicted noise. I would like to note that this formalism suggests that the individual samples in the trajectories are independent from each other. ie. a change in one sample does not influence the probability of another sample in the same trajectory. Given that we are looking at trajectories, this is not true. In other words, the inductive bias introduced by the authors does not hold for the modelling of trajectories. This is a breaking issue and I would like to get the authors view on this observation.

**Experiments**
1. For the discussion of the datasets obtained, it is unclear how the long trajectory is divided into shorter trajectories for the training set.
2. Regarding the referencing to figures, please correct this to use the correct referencing format where it reads as either "fig. x" of "figure x" instead of simply the number.
3. Regarding the discussion of the Brownian Motion in a Benchmark potential correctly sampling the wells, the conditional sampling changing the spread of the samples, and correctly identifying the Boltzmann distribution, I'm unsure if these results are informative. As it stands, given that the training data also correctly covers the Boltzmann distribution, it is not surprising that the proposed diffusion method can correctly sample this space. The issue with rare event sampling is that the temperature of the simulation is too low to effectively sample transitions and as a result, the ratio in Boltzmann distribution between different states is incorrect. The temperature for generating the dataset however seem to be high enough to already get a correct estimate. This is a breaking issue and I would appreciate the authors comments on this.
4. Additionally, an important aspect of the papers contribution is that they extend diffusion models for molecular conformations to the study of dynamics. As of now, the presented evaluation however has focussed on properties of molecular systems that can also be studied without taking into account the dynamics. Standard diffusions models should, in theory, be able to sample the Boltzmann distribution and provide accurate estimates of the FES. This is an breaking issue and I would like to see a more in depth study of the dynamical properties that can not be studied using standard diffusion models.
5. In figure 4, the description of (b1) and (b2) mention Free Energy profiles but the figure shows equilibrium distributions instead.
6. For 3.2 the authors state, "a (relatively) rare conformational transition occurs around the dihedral angle $\psi$". However, the energy barrier between these two states is considered to be relative low and can thus be sampled in relative short periods at low temperatures. As such, I'm unsure if this transition is suitable for evaluating the method. I would suggest to instead consider transition along the $\phi$ axis.
7. Again, given that the training for Alanine Dipeptie already correctly determines the FES I'm unsure what the added benefit is from modelling the transitions using a diffusion model. The FES can also be reconstructed using a sample based diffusion model.
8. The last sentence of section 3.2 reads "pointing to the generalisation capabilities of the method" regarding the observation of a rare transition along the $\psi$ angle. I respectfully disagree with the authors here. This transition is highly physically unrealistic as it flows over a region of high energy.
9. Regarding the ablation study, based on the presented error bars these results are all not significant. This is a breaking issue and I would like the authors comments on how this affects their results

**Conclusion**
1. The authors state "which can serve as a useful surrogate model of the physics-based simulator, due to its enhanced sampling capabilities". Aside from possible speedup, the presented results do not currently show any enhanced sampling capabilities. As such, I think it would be appropriate for the authors to rephrase this claim.
2. The authors discuss the extension of their work to generalise accross single molecules. I believe this to be a very good suggestion and hope that the authors continue their research in this direction.

**References**

[1] https://pubs.acs.org/doi/10.1021/acs.chemrev.0c01111

[2] https://pubs.acs.org/doi/10.1021/acscentsci.8b00913

[3] Differentiable Simulations for Enhanced Sampling of Rare Events

[4] Learning Free Energy Pathways through Reinforcement Learning of Adaptive Steered Molecular Dynamics

[5] https://arxiv.org/abs/2207.02149

[6] https://iopscience.iop.org/article/10.1088/2632-2153/acf55c (Only recently published but possibly relevant for the authors)

---

### Official Review · Reviewer_BCdf · 2023-10-30

**Soundness:** 2 fair
**Presentation:** 1 poor
**Contribution:** 3 good
**Rating:** 3
**Confidence:** 4

**Summary:**

The paper proposed Dynamics Diffusion which is based on the denoising diffusion probabilistic model to enable the generation of molecular dynamics simulation trajectories of molecules and the sampling of rare events in the MD simulation. The Dyanmics Diffusion is used to train a UNet-based model which takes in the 3D coordinates of atoms and outputs a single value of conformation energy, of which the Jacobian can be calculated as the predicted noise. The model can also be conditioned on extra parameters such as the simulation temperature to enable conditional generation. To achieve rare event sampling of MD simulation trajectories, the authors adopted the methods of impainting and variation generation that condition the model on specific frames of the trajectories. Experiments have been done using a self-created dataset of particle Brownian motion in double-well and the public alanine dipeptide MD trajectory. Ablation studies of this paper focuses on the UNet architecture (encoder only, self-attention, etc.)

**Strengths:**

1. Despite some prior work in using diffusion model to train neural networks for the purpose of sampling MD trajectories, application of such method in rare-event sampling in low probability density region is still an under-research field. The impainting and variation generation methods used in this work are very interesting ideas which are demonstrated to work in the paper.

2. The formulation of the Dynamics Diffusion that predicts the total energy of each frame is similar to methods used in many machine learning potential works. The forces on atoms can be then calculated as the gradient of energy by taking the Jacobian. Such a method ensure the physical validity between the force on atoms and the energy of system.

**Weaknesses:**

1. The transferability of the proposed UNet architecture can be problematic. The UNet takes a vector with the dimension of $3N$ as input and outputs a scaler (sum of energy of all $N$ atoms). The number of atoms is fixed and there is no information about the atom type. Such design means that the model after training is dedicated to a specific molecule without the possibility of being transferred to another system. It also means that for each molecule, a dataset of molecular dynamics simulation trajectories has to be curated for model training. Although the model can still be used for rare-event sampling as the probability of rare-events can be too low to be caught in MD simulation, the value in generating full MD simulation trajectory using such a model is low.

2. The number of system tested in this paper is limited. The particle in double-well and alanine dipeptide are relatively small systems. Large systems such as protein dynamics (www.science.org/doi/10.1126/science.1208351) can be added to further strength the claim in this work.

3. The presentation of this work has plenty of room for improvement. The resolution of figures are low (especially Fig. 8 and 11). Labels of both axes in Fig. 11 are illegible. Model related information (architecture, optimizer etc) for both double-well and alanine dipeptide systems are highly redundant. No parentheses or brackets are used for reference in the manuscript. Reading the manuscript with reference author names randomly seperating the sentence is a great experience. When referring to a figure or table, the word "Figure" or "Table" is sometimes missing. Latex render error occurs in page 18 "Result" section.

4. Some related works of learning MD trajectories using diffusion models (arxiv.org/abs/2305.18046) and rare event sampling using differentiable simulation (arxiv.org/abs/2301.03480) are missing from literature review.

**Questions:**

1. As shown by the authors, taking Jacobian of the predicted energy term can be costly than predicting the noise (Table A.1). However, the authors only benchmarked the error of models (predicting energy or force) in the 2D double-well system. It is not convincing enough to me that energy prediction+Jacobian is better than force prediction in terms of cost/accuracy tradeoff for larger systems.

2. In Figure 4.c, the trajectories generated by inpainting methods seems to capture the molecular transition dynamics between the two conditioned points. However, the left condition (green dot) is not connected to other states. Is that a plotting mistake?

3. How is the self-attention implemented along with the UNet architecture? The detail is missing in the manuscript.

4. The model architecture and problem formulation enables rare-event sampling with a fixed number of timestep ($\tau$) when the initial and end conditions are given. Based on my understand, the method can garuantee the transition states between the two condition to be sampled. However, the probability of those rare-events cannot be obtained using the model.

---

### Official Review · Reviewer_2zYs · 2023-10-31

**Soundness:** 2 fair
**Presentation:** 2 fair
**Contribution:** 1 poor
**Rating:** 3
**Confidence:** 5

**Summary:**

This paper proposes DynamicsDiffusion, a method using the denoising diffusion probabilistic model to simulate the trajectories of molecular dynamics.

**Strengths:**

- This paper studies an important problem of MD simulation.
- The qualitative results are nice.

**Weaknesses:**

Unfortunately, the key idea of this paper, using generative modeling (DDPM) for MD simulation, has been done in several works before [1, 2, 3]. They have been archived for over six months, and all have been accepted to relevant conferences/journals. The methods have almost no difference.

[1] https://pubs.acs.org/doi/10.1021/acs.jctc.3c00702

[2] https://dl.acm.org/doi/abs/10.1609/aaai.v37i4.25663

[3] https://openreview.net/forum?id=y8RZoPjEUl

**Questions:**

N/A

---

> ### Public Comment · ~Song_Liu10 · 2023-11-12
> **Clarifying the Novel Focus of DynamicsDiffusion on Rare Event Sampling in All-Atom MD Simulations**
>
> As an observer and not the author of "DynamicsDiffusion: Generating and Rare Event Sampling of Molecular Dynamic Trajectories Using Diffusion Models," I'd like to share some thoughts on Reviewer 2zYs's assessment. While the reviewer aptly recognizes the paper's significance and the strength of its qualitative results, there appears to be a slight misinterpretation regarding its primary focus.
>
> Speaking from a computational biologist's perspective, the unique contribution of DynamicsDiffusion lies in its targeted approach to the complex challenge of rare event sampling in all-atom molecular dynamics (MD) simulations, utilizing denoising diffusion probabilistic models (DDPM). This focus is revolutionary, as sampling these rare events, which are integral in analyzing transition states, is an extremely challenging yet essential part of MD simulations. These rare events are key to unraveling the functional mechanisms of molecular systems. DynamicsDiffusion's nuanced application of DDPM to tackle this specific challenge marks a significant leap forward, potentially reshaping our comprehensive understanding of molecular dynamics, particularly in systems where transition states are crucial.
>
> I'd like to respectfully point out what seems to be a slight oversight in the review concerning the differentiation between all-atom MD simulations and Coarse-Grained MD simulations. The studies cited in the review largely pertain to diffusion model applications in Coarse-Grained MD and enhancing MD simulations by directly forecasting 3D coordinates. However, DynamicsDiffusion sets itself apart by concentrating on rare event sampling within all-atom MD simulations. This distinction is paramount, given that all-atom simulations offer a more intricate and detailed portrayal of molecular interactions, essential for accurately capturing and analyzing these rare events. Such subtleties are often not given due attention, underscoring the necessity for a more in-depth understanding within the field. Therefore, DynamicsDiffusion's specific focus on this nuanced aspect of MD simulations represents an important advancement, deserving of recognition and further consideration for its distinct contribution to molecular dynamics at the granular, all-atom level.
>
> I appreciate the opportunity to contribute to this discussion.

---

> > ### Author Response · Authors · 2023-11-21
> >
> > Dear Song Liu,
> > Thank you for your insightful and clarifying remarks on our paper! Your kindness in sharing your expertise is truly appreciated. It means a lot to me.
> > Warm regards,
> > Author1

---

> ### Comment · Reviewer_2zYs · 2023-11-12
> **Response**
>
> Hi there,
>
> Thank you for initiating this discussion. I am very happy to share more thoughts.
>
> First, about my prior comments, let me expand more details on my expectations:
> 1. The problem is interesting, and this is not just a compliment.
> 2. But, the method is too weak. There is a nine-page limitation, while less than one page (half of Sec 2.2 and Sec 2.3) is about the method, and their main idea has been scooped. Besides, these prior works (DDPM for MD) are quite well-known, and the authors missed the whole core research line. What I expect the authors to reply during the rebuttal is to dig further into the method, and tell the audience (1) how their ML method is different from the other works; (2) existing works are about simply applying DDPM to MD, so how this work is specific in generalizing to handle the rare-event sampling under Brownian dynamics; (3) if there's no fundamental difference, then does that mean we can adopt these existing DDPM for MD method and apply on this paper's dataset? These are the questions I left to the authors during the rebuttal in the first-round comments.
> 3. There are some minor issues with the dataset/problem setting, on why Brownian dynamics where the inertial force is ignored and not other forms of Langevin dynamics, the NVE/NVT/NPT configuration, in the vacuum or in the gas/solvent molecules, etc. I am not sure if other reviewers have pointed this out. But as an ML paper, even if the problem setting is not perfect, I am fine with it, as long as it has some ML merit.
> 4. The reason I only put one weakness point, is because I know this field, and this is the main thing that needs to be answered. If the authors can answer this, I will raise the score to borderline at least.
>
> Then to respond to your comments:
> 1. According to your descriptions, you also show that the main merit of this paper is not about the method/pipeline/framework, but about the problem setting. In general, everyone can claim that their problems are important, but that doesn't mean adopting an existing method on such a problem without any modification can be a top-tier ML work.
> 3. There are many implementation details in DDPM for MD, and switching between coarse-grained and all-atom simulations is a trivial thing as long as the GPU memory holds (feel free to check the DDPM for MD papers listed above). Besides, the all-atom modeling is adopting previous work.
>
> ---
> To sum up, I wrote a concise and critical review, and I expand more details here. But indeed, they are all covered in my first-round review: `What's the difference`. I am even happy to defend this work if the authors can answer my question.
>
> I also want to point out that, writing a concise and critical review doesn't mean irresponsible, but writing sth. nonsense is. Let me list a few things that mark an irresponsible review:
> - Tell the authors that they miss important baselines without giving the references.
> - Tell the authors to add the reviewer's works as baselines, which are actually not relevant.
> - Ask the authors open questions that cannot be solvable at the current stage, like "a DL model lacks interpretation".
> - The reviewer completely misses the math, yet insists that the authors are wrong.

---

### Official Review · Reviewer_njs8 · 2023-10-31

**Soundness:** 3 good
**Presentation:** 1 poor
**Contribution:** 2 fair
**Rating:** 3
**Confidence:** 4

**Summary:**

Based on DDPM, the author proposes a extension of dynamics diffusion to generate molecular dynamics. The model leverages the conditional generation property of DDPM to generate various trajectories. The model is the first deep generative model for molecular dynamics to the authors' knowledge.

**Strengths:**

Originality: Combination of DDPM and molecular dynamics. The original part of DDPM is to introduce two variants of sampling. Moreover, a physics-inform layer is introduced to the Unet block.

Quality: A lot of essential parts are missing.

Clarity: The writing is poor. Many typos and the weird reference style make it hard to read smoothly. There are also missing references in the appendix.

Significance: It is the first generative model for molecular dynamics. The approach enables rare events enhanced sampling. It can reconstruct free energy and dynamics accurately.

**Weaknesses:**

The novelty seems straightforward compared with the original DDPM. The reference style looks weird. The writing is not organized. Some important details are missing. For example, what is the math expression of the energy formulation? The paper also misses many key results to support the claims. What is the performance difference between your proposed 2 sampling methods and the baseline method? Also, the authors claim the proposed model is suitable for the symmetries, but the reviewer didn't find the corresponding result to support it. There is no part to cite Figure 2, which makes the reviewer guess which part corresponds to Figure 2.

**Questions:**

1. What is the math expression of the energy formulation?
2. What is the performance difference between your proposed 2 sampling methods and the baseline method?
3. Which part is related to Figure 2? The reviewer can guess but wants it to be precise.
4. Wrong reference table for Page 8. The author cited Table 9 for the ablation study of self-attention. But Table 9 is about Jacobian. Missing references on page 18.
5. Which part of the result shows it can promote symmetry?
Miscellaneous typos need to be further proofread.

---

### Author Response · Authors · 2023-11-21

Dear Reviewers,
We appreciate the opportunity to provide further clarification on our paper.

Clarification of Novelty and Contribution:

Our paper, "DynamicsDiffusion: Generating and Rare Event Sampling of Molecular Dynamic Trajectories Using Diffusion Models," introduces an innovative approach to utilizing denoising diffusion probabilistic models (DDPM) to address the challenge of sampling rare events in all-atom molecular dynamics (MD) simulations. These rare events are crucial for understanding the functional mechanisms of molecular systems, especially in analyzing transition states. Unlike previous studies, mentioned by one reviewer, focused on Coarse-Grained MD simulations [1], generating just conformations [1], or speeding up sampling [2][3], our work generates trajectories, by including the time dimension, of full all atom systems. We believe this application of DDPMs could offer a valuable tool for MD practitioners in the future. We acknowledge that there are other shortcomings but we believe that this work is truly novel and not covered by existing work.

Acknowledgment of Criticism:

We want to acknowledge that the UNet architecture is likely not ideal, as it can't be generalized beyond one system, and a GNN-based architecture might be more appropriate. Furthermore, more experiments and visualizations showing individual-generated trajectories should be added to the paper to show that the model is capable of generating such trajectories. The model also has issues generating trajectories that are longer than the length of the trajectories, up to 5120 simulator steps, which might cause issues for some applications. Lastly, we want to thank the reviewers for pointing out points that are not presented, missing citations, and other not methodological points. We will try to address these points in the next iteration of the paper.

Importance of Our Contribution in the Conference Context:

We believe that papers like and similar to ours would significantly contribute to the interdisciplinary dialogue between deep learning and natural sciences at ICLR. Our work exemplifies the potential of deep learning in addressing complex scientific challenges, particularly in the nuanced field of molecular dynamics where there are a host of challenges to be addressed. Although we are saddened to see the reception, we will not withdraw the submission for the reviews and our comments to remain visible.

We also extend our gratitude to Song Liu for his insightful and supportive comment on the review.

[1] https://pubs.acs.org/doi/10.1021/acs.jctc.3c00702
[2] https://dl.acm.org/doi/abs/10.1609/aaai.v37i4.25663
[3] https://openreview.net/forum?id=y8RZoPjEUl

---

### Meta-Review · Area_Chair_tmML · 2023-12-04

**Metareview:**

The paper presents an approach using DDPM to generate molecular dynamics simulation trajectories, with a focus on rare event sampling. The manuscript was reviewed by five reviewers and all unanimously agreed on rejection. Authors responded to the reviews. However, many major issues still remain. Specifically:

- Methodological Novelty: The proposed method (from a machine learning algorithm perspective) is not really novel and similar methods have been utilized fro similar molecular dynamics simulation problems.

- Limited System Testing and Transferability Issues: The experiments are limited to small systems, and there is a lack of evidence demonstrating the generalizability and scalability of the proposed method to larger, more complex systems. Additionally, the UNet architecture's transferability appears limited, which raises concerns about the practical applicability of the proposed method.

In summary, it is clear that the paper lacks methodological novelty. This is not a ground for rejection as the conference welcomes application papers and the application area is an important one. However, the paper writing focuses much more on methodological novelty instead of focusing on target application. Even the impact on the proposed domain is not demonstrated confidently. Overall, I strongly suggest authors to rewrite the paper focusing entirely on the application and provide further experiment showing transferability and the impact on the domain of interest.

**Justification For Why Not Higher Score:**

The manuscript is not ready for publication and would benefit significantly from a major re-write.

**Justification For Why Not Lower Score:**

N/A

---

### Decision · Program_Chairs · 2024-01-16

Reject